# Thermal Modification Effect on Supported Cu-Based Activated Carbon Catalyst in Hydrogenolysis of Glycerol

**DOI:** 10.3390/ma13030603

**Published:** 2020-01-29

**Authors:** Juan Seguel, Rafael García, Ricardo José Chimentão, José Luis García-Fierro, I. Tyrone Ghampson, Néstor Escalona, Catherine Sepúlveda

**Affiliations:** 1Facultad de Ciencias Químicas, Universidad de Concepción, Edmundo Larenas 129, Casilla 160C, Chile; juanseguelr@udec.cl (J.S.); rgarcia@udec.cl (R.G.); rchimenton@udec.cl (R.J.C.); 2Millenium Nuclei on Catalytic Processes towards Sustainable Chemistry (CSC), Santiago 7810000, Chile; neescalona@ing.puc.cl; 3Instituto de Catálisis y Petroleoquímica, CSIC, Cantoblanco, 28049 Madrid, Spain; jlgfierro@icp.csic.es; 4Department of Chemical System Engineering, The University of Tokyo, 7-3-1 Hongo, Bunkyo-ku, Tokyo 113-8656, Japan; isghampson@uc.cl; 5Departamento de Ingeniería Química y Bioprocesos, Escuela de Ingeniería, Pontificia Universidad Católica de Chile, Avenida Vicuña Mackenna 4860, Macul, Santiago 7810000, Chile; 6Facultad de Química y de Farmacia, Pontificia Universidad Católica de Chile, Santiago 7810000, Chile; 7Unidad de Desarrollo Tecnológico, Universidad de Concepción, Coronel 4190000, Chile

**Keywords:** glycerol, copper, activated carbon

## Abstract

Glycerol hydrogenolysis to 1,2-propanediol (1,2-PDO) was performed over activated carbon supported copper-based catalysts. The catalysts were prepared by impregnation using a pristine carbon support and thermally-treated carbon supports (450, 600, 750, and 1000 °C). The final hydrogen adsorption capacity, porous structure, and total acidity of the catalysts were found to be important descriptors to understand catalytic performance. Oxygen surface groups on the support controlled copper dispersion by modifying acidic and adsorption properties. The amount of oxygen species of thermally modified carbon supports was also found to be a function of its specific surface area. Carbon supports with high specific surface areas contained large amount of oxygen surface species, inducing homogeneous distribution of Cu species on the carbon support during impregnation. The oxygen surface groups likely acted as anchorage centers, whereby the more stable oxygen surface groups after the reduction treatment produced an increase in the interaction of the copper species with the carbon support, and determined catalytic performances.

## 1. Introduction

Glycerol is produced in large amounts as a reaction by-product during biodiesel production [1,2]. Glycerol has been identified by the Department of Energy of United States (DOE) as one of the top-12 building block chemicals that can be derived from sugar and converted to high-value bio-based chemicals or materials [3]. Glycerol is a non-toxic, edible, bio-sustainable, and biodegradable compound [4]. Availability and low price of glycerol makes it a promising feedstock for producing a wide range of value-added chemicals. 

The use of glycerol for production of chemicals is of great industrial importance. For example, propanediols such as 1,2 propanediol (1,2-PDO) is an important commodity chemical which can be used as polyester resins, liquid detergents, pharmaceuticals, cosmetics, and animal feed [5]. 1,2-PDO is currently produced from petroleum derivatives such as propylene oxide [6]. Alternative routes to the formation of 1,2-PDO can be traced from renewable feedstocks, involving for example, hydrogenolysis of glycerol.

Several research works have used heterogeneous catalysts such as Zn, Cu, Mg, Co, Mo, Pd, Ni, and Pt to catalyze the hydrogenolysis of glycerol. The main products of this reduction process were ethylene glycol, 1,2 propanediol, 1,3 propanediol, lactic acid, acetol, propanol, or even acrolein. The reactions have been conducted within a range of temperature (473–623 K) and pressure (2000–5000 psi). Conventional hydrogenation catalysts based on nickel, ruthenium and palladium are not effective when employed for catalytic -OH cleavage. Copper has been extensively reported as a selective active phase for the production of propylene glycol [7]. Hydrogenolysis of two primary hydroxyl groups yields 1,2-PDO, 2-propanol, and eventually propane. The primary or secondary hydroxyl groups of glycerol are more easily reduced depending on the catalytic system and reaction conditions. Montassier et al. [8] proposed a reaction mechanism for the conversion of glycerol to 1,2-PDO. In this mechanism, the cleavage of C-C bond is proposed to occur through a base-catalyzed retro-aldol reaction, whereas C–O cleavage occurs through base-catalyzed dehydration reaction. Suppes et al. [9] proposed that C–O cleavage leading to the formation of propylene glycol is a metal-catalyzed reaction. In this route, glycerol is directly dehydrated on the metal site to form acetol as an intermediate, which can then be hydrogenated over the metal to form propylene glycol (1,2-PDO). Copper is known for its poor hydrogenolytic activity towards C-C bond but it is an efficient catalyst for C-O bond hydrogenation and dehydrogenation reactions. In fact, different publications using Cu as a metallic active phase on different supports (ZrO_2_, Al_2_O_3_, ZSM-5, ZnO, SiO_2_) have shown 1,2-PDO as a principal product [10,11,12,13,14].

Literature studies have shown that a variety of catalysts—such as ion exchange resins, sulphates, phosphates [1], heteropolyacids supported on silica, alumina [15], and zeolites [16]—have been employed to transform glycerol. Catalysts used also include acidic ZSM-5, beta, SAPO, and Faujasite supported on heteropolyacids [17,18]. Glycerol hydrogenolysis using copper supported on ZnO (473 K, 42 bar, 24 h) has been reported, although the selectivity to propanediols was low [19]. In addition, previous research has shown that silica supports are not suitable for aqueous-phase processes because silica is hydrolyzed under reaction conditions leading to coalescence and sintering of metal particles on the support [20]. Conversion of glycerol to propanediols may proceed by the combination of dehydration over acid sites with subsequent hydrogenation over metallic sites [21]. Previous research has shown that hydrogenolysis of glycerol to 1,2-PDO proceeds via dehydration of glycerol to acetol on acid catalysts and consecutive hydrogenation over metal catalysts. Solid acid catalysts have been reported to enhance the conversion and selectivity in the hydrogenolysis of glycerol [22,23]. Carbon materials satisfy most of the desirable requirements for suitable catalyst support for glycerol hydrogenolysis due to its high surface area and optimum porosity [24]. Sufficiently high surface area and optimum porosity are essential to provide high accessibility to the active sites of the catalyst. The chemical reactivity of carbons is due primarily to the existence of unsaturated valences at the edges of graphite-like hexagonal crystallites. Metal interaction can be considerably enhanced by the introduction of acidic functional groups through surface oxidation using different agents such as nitric acid, hydrogen peroxide, etc. Oxidized active carbons possess some unique properties mainly due to the considerable amount of oxygen-containing surface functional groups [25]. It has been pointed out that the more thermally stable oxygen surface groups are more effective in anchoring metal species and increasing its dispersion. The metallic dispersion on carbon support has been linked to porous structure of the activated carbon [26], the oxygen groups present on activated carbons [27] and specific interactions of metal precursors with surface defects [28].

Although Cu has been used as an active phase on several support with different physicochemical properties [10,11,12,13,14], its use on activated carbon has only been as a promoter of Pt for glycerol oxidation [29]. Therefore, the present work investigates the effect of thermal treatment of activated carbon as supports for copper catalysts for glycerol conversion. The thermal treatment influences the amount of oxygen surface groups, which consequently controls acidity and adsorption capacity of the copper catalyst. It is expected that oxygen surface groups will have a crucial influence on the final accessibility to the exposed copper species on the catalyst. 

## 2. Materials and Methods 

### 2.1. Catalyst Preparation

The Cu supported on activated carbon (CGran) catalysts, denoted as Cu/CGran(X) catalysts, were prepared by wetness impregnation using aqueous solution of Cu(NO_3_)_2_·3H_2_O (Sigma Aldrich, Darmstadt, Hesse, Germany) as precursor salts and CGran Norit commercial activated carbon as support. The activated carbon support was previously crushed and sieved to 80 and 125 μm particle size, and modified by thermal treatment prior to impregnation. The thermal treatment was carried out in an inert environment (He) at four different temperatures (450, 600, 750, and 1000 °C). The temperature program was from 25 °C to the modification temperature indicated (heating rate of 10 °C min^−1^), which was then maintained for 1 h. The supports were correspondingly denoted as: CGran (450), CGran (600), CGran (750), and CGran (1000). Then, catalysts with copper loading of 7 wt.% were prepared on un-modified and modified CGran activated carbon supports (five Cu/Gran(x) catalysts), left at room temperature for 24 h, and finally dried at 120 °C for 12 h. Before the reaction, the catalysts were reduced ex-situ at 350 °C for 4 h in flowing H_2_ (60 mL min^−1^). The Cu contents were determined by atomic absorption spectroscopy (AAS) using a Thermo Scientific ICE 3000 model series (Waltham, MA, USA) 

### 2.2. Characterization of Supports and Catalysts 

BET surface area (S_BET_), total pore volume (V_p_), mesopore volume (V_m_), and micropore volume (V_0_) were determined by nitrogen sorption measurement at −196 °C using Micrometrics TriStar II 3020 equipment (Beijing, China). Prior to the measurements, the samples were outgassed at 300 °C for 2 h. The BET surface area was calculated from the adsorption branch of the isotherms in the range 0.05 ≤ *P/P°* ≤ 0.15 while the total pore volume was recorded at *P/P^0^* = 0.99. 

Temperature programmed decomposition–mass spectrometry (TPD–MS) was utilized to determine of oxygenated surface functional groups on the activated carbons. CO and CO_2_ TPD profiles were obtained with Micromeritics Autochem II equipment (Beijing, China). 100 mg of sample was packed in a U-shaped quartz tube and placed inside a furnace. The sample was heated to 1050 °C, with a heating rate of 10 °C min^−1^ under a helium flow of 50 mL min^−1^. A quadrupole mass spectrometer Cirrus 2 was used to monitor CO and CO_2_ signals. Fourier-transform infrared (FTIR) spectroscopy of the supports were performed on a Nicolet Nexus (Waltham, Massachusetts, United States) FT-IR in the middle range (4000–400 cm^−1^) and averaged after 64 scans. The tablets were prepared using KBr as a support in a 1:100 mg proportion of sample to support.

The total acidity measurements of the carbon supports and catalysts were carried out potentiometrically by titrating a suspension of carbon in acetonitrile with n-butylamine using an Ag/AgCl electrode [30].

Temperature programmed reduction (TPR) of the calcined sample was carried out in a quartz cell using a Micromeritics AutoChem II equipment equipped with a thermal conductivity detector. Approximately, 20 mg of the sample was heated under 5% H_2_/Ar flowing at 50 m Lmin^−1^, and heated from 25 °C to 1050 °C (10 °C min^−1^). The consumption of hydrogen was determined after the internal calibration of the TCD.

N_2_O reactive chemisorption was performed using the same Micromeritics AutoChem II device. A quantity of about 80 mg of catalyst was introduced into the fixed bed tubular reactor. The N_2_O consumption was calculated from a two-step analysis consisting of (i) N_2_O oxidation of Cu to Cu_2_O (2Cu + N_2_O → Cu_2_O + N_2_) and (ii) hydrogen temperature-programmed –reduction of the formed Cu_2_O surface species. Briefly, the supported catalysts were reduced in 5%H_2_/Ar (20 mL min^−1^) at 5 °C/min from 35 to 350 °C, and the temperature was held at 350 °C for 4 h. Selective oxidation of the copper surface to Cu_2_O was performed under 20% N_2_O/Ar flow (20 mL min^−1^) at 60 °C. Then, Cu_2_O surface was reduced with 5% H_2_/Ar (20 mL min^−1^) at 5 °C min^−1^ from 35 to 900 °C (H_2_ + Cu_2_O_surface_ → 2Cu_surface_ + H_2_O). Finally, the total hydrogen consumption of the surface selectively oxidized by N_2_O was performed with the internal calibration of the TCD.

XPS measurements were performed using a VG Escalab 200R electron spectrometer equipped with a hemispherical electron analyzer and Mg Kα (1253.6 eV) X-ray source, according to procedure detailed in our previous publication [14]. 

### 2.3. Catalytic Tests

Hydrogenolysis of glycerol was performed according to detailed experimental conditions reported in our previous work [14]. The condition of the reaction included an aqueous glycerol solution (80 wt % or 10 mol L^−1^), a 0.5 g of pre-reduced catalyst, a temperature of 220 °C and a total pressure of 5 MPa of H_2_.

The initial rate (r_0_) was expressed in mol of glycerol per gram of catalyst per unit time (molg_cat_^-1^ s^-1^) and calculated according to Equation (1): (1)$$r_{0}=\frac{b\;x\;n^{0\;}Gly}{m}\;\;\;\;\;\;\;$$
where *b* is the initial slope of the conversion, n^0^ Gly is the initial mol of glycerol and, *m* is mass of catalyst.

The intrinsic rate was calculated from the specific rate according to the following equation:(2)$$r_{1}=r_{s}\frac{N_{av}}{nCu}$$
where r_1_ is the intrinsic rate (molecules of glycerol transformed per Cu atom per second), r_S_ is the specific rate (moles of glycerol transformed per gram of catalyst per second), nCu represents the total number of Cu atoms per gram of catalyst, and N_av_ is Avogadro’s number. The product distribution was calculated when the reaction was completed.

## 3. Results

### 3.1. Catalysts and Support Characterization

#### 3.1.1. Temperature Programmed Decomposition–Mass Spectrometry (TPD–MS)

Differences in the chemical nature of the oxygen functional groups of un-modified and modified CGran activated carbon used as supports were determined by means of TPD profiles, as shown in Figure 1.

Figure 1a shows that un-modified activated carbon (AC) support presents a shoulder at low temperature (200–400 °C), an intense peak at intermediate to high temperature (500–800 °C), and a weak signal at high temperature (700–900 °C). According to Figuereido et al. [31], the peaks can be assigned to different oxygen surface groups: lactonic groups (200–700 °C) [32,33]; carboxylic (200–300 °C) [32]; phenolic (600–700 °C) [32]; carbonyls (900–1000 °C) [32]; and quinones (700–1000 °C) [33]. The decomposition of these groups can be associated with the release of CO_2_ or CO, as confirmed by the TPD-MS profiles in Figure 1b for un-modified CGran AC. This result suggests the presence of acidic groups, consistent with previous results which found strong acid sites on CGran AC by TPD profiles coupled to a non-dispersive infrared device [34]. Figure 1 also shows that thermal treatment affects the intensities of the peaks, and correspondingly, the nature and amount of surface oxygen groups. A selective decrease of oxygen functional groups can be observed depending on the temperature of modification: for CGran (450), carboxylic groups decreased, whereas for the other modified AC supports, quinonic groups decreased in the order Gran (600) > CGran (750) > CGran (1000). These TPD results indicate that selective removal of functional groups occurs as the treatment temperature increased, as expected.

#### 3.1.2. Fourier Transformed Infrared Spectroscopy (FT-IR)

Figure 2 shows the FT-IR spectra of the modified CGran(x) support compared with CGran AC [34]. The characteristic bands of these solids are in the region of 3500–3300 cm^−1^, assigned to phenolic structures; 1600–1500 cm^−1^, ascribed to C=O stretching vibrations of lactonic, quinonic, or carboxylic groups, and 1300–900 cm^−1^, assigned to C–O–C vibrations or C–O stretching of ether, lactonic carboxylic, or phenolic groups [34,35]. In addition, Figure 2 shows a decrease of all the oxygen functional groups with the increase of the temperature of the thermal treatment, which is in agreement with results deduced from TPD profiles.

#### 3.1.3. Total Potentiometric Acidity

The acid strength determined by potentiometric titration and expressed as total acidity (E_0_) is summarized in Table 1. All catalysts were reduced, following the same protocol employed for the activation of the catalysts prior to the catalytic tests. For comparison, potentiometric titration of un-modified CGran AC was included. Strength of acid sites can be classified according to a scale reported by Cid and Pecchi [30]: E_0_ > 100 mV (very strong acid sites), 0 < E_0_ < 100 mV (strong acid sites), −100 < E_0_ < 0 mV (weak acid sites), and E_0_ < −100 mV (very weak acid sites). As expected, the CGran AC presents very strong acid sites, in agreement with TPD results. Table 1 shows that the E_0_ of the reduced catalysts decreases with the increases of the temperature of thermal treatment. This behavior could be attributed to the deposition of copper species over oxygen surface groups of the activated carbon support together with decrease of oxygen functional groups by thermal treatment [26]. 

#### 3.1.4. Nitrogen Sorption at 77 K

The textural properties of the un-modified and modified CGran activated carbons, and the corresponding Cu/CGran(x) catalysts obtained from nitrogen sorption isotherms are shown in Figure 3 and summarized in Table 1.

Figure 3 shows that the CGran and CGran(x) supports exhibit the same behavior, namely a high initial adsorbed volume, characteristic of microporous materials. In addition, at high relative pressures, an increase in the adsorbed volume is observed, indicating the presence of mesoporous structure. All solids displayed a type IV isotherm [36] and their hysteresis loop corresponds to a type H4, suggesting that the supports are mainly mesoporous with thin slit of pores [36]. Additionally, as the modification temperature increases, the porous volume of the support decreases, indicating that the heat treatment produces a change in the textural properties of the activated carbon. In this context, Table 1 shows that thermal modification slightly modified the CGran (450) sample, in comparison to the un-modified carbon support. For the CGran (600) and CGran (750) samples, their apparent surface area (S_BET_) and mesoporous volume (V_m_) decreased by 14 %, while the microporous volume (V_o_) diminished by 20%. Textural properties of CGran (1000) sample was the most compromised: the high temperature treatment likely promoted annealing of the grains of the carbon support, resulting in a decrease in the apparent surface area [37]. After Cu impregnation, the S_BET_ values decreased by 38% for the Cu/Gran catalyst, and 27–37% for the Cu/CGran(x) catalysts. The V_m_ and V_o_ values decreased by 44% and 51% for Cu/Gran catalyst, respectively. Similarly, these values decreased by a similar degree for the Cu/CGran(x) catalysts. The slight decrease of textural properties among the Cu/CGran(x) catalysts compared to the un-modified Cu/CGran catalyst can be attributed to a more homogeneous distribution of copper precursor due to elimination of hydrophobic oxygen species from the surface of CGran support. This hydrophobic character and its effect on textural properties was proposed in a previous work [34], and similar results using activated carbon was obtained by Lagos et al. [38].

#### 3.1.5. Temperature Programmed Reduction (TPR)

Figure 4 shows the TPR profiles of Cu/CGran and Cu/CGran(x) catalysts. All the catalysts displayed one prominent reduction peak, observed around 270 °C for Cu/CGran, Cu/CGran (450) and Cu/CGran (1000) catalysts, and 350 °C for Cu/CGran (600) and Cu/CGran (750) samples.

The principal peaks at 270 °C and 350 °C are assigned to the reduction of dispersed CuO species to Cu [39,40]. The shift in the reduction temperature peak to a higher temperature can be attributed to the elimination of hydrophobic surface groups and the permanence of hydrophilic oxygen surface groups, such as quinones, after thermal treatment at 600 °C and 750 °C. This hydrophilic character in CGran (600) and CGran (750) samples favors the homogeneous distribution of Cu species [34,38], resulting in an increase in the metal-support interaction. The homogeneous distribution of copper species may contribute to the observed increase in reducibility as shown Table 1.

On the other hand, the lower reduction temperature peak observed for the Cu/CGran (1000) catalyst is due to a decrease of the interaction between the metal and carbon support due to the near total elimination of the oxygen functional groups. Figure 4 also shows a broad feature between 450 and 700 °C for all catalysts, which progressively disappears as the temperature of the thermal treatment increases. This shoulder can be attributed to the consumption of hydrogen by the oxygen groups of the corresponding activated carbon surface, methane generation, or organic compounds decomposition producing CO by surface support gasification [32].

Table 1 shows the reducibility (%) of Cu/CGran and Cu/CGran(x) catalysts calculated from TPR measurements. The reducibility increases as the temperature of the thermal treatment increases up to 750 °C. This behavior can be related to the presence of thermally-stable oxygen species on the AC support which anchors and increases Cu species on the surface, and as a result increases copper reducibility (%). On the contrary, the reducibility of the Cu/CGran(1000) catalyst is the lowest at 81 %, probably due to a decrease of the interaction of Cu species with the support as a result of loss of stable thermal oxygen groups. 

#### 3.1.6. N_2_O Titration (N_2_O-TPR)

One important feature of N_2_O titration of copper catalysts is the total quantity of H_2_ consumed (expressed in μmol · g_cat_^-1^) measured by temperature-programmed reduction (TPR) of the selectively oxidized surface copper atoms by N_2_O (Table 2). 

Assuming that hydrogen consumption is only modulated by the dispersion of copper on the catalyst surface, Table 2 shows the amount of Cu active sites calculated from experimental N_2_O-TPR. However, if these results are compared with the total Cu content in the catalyst obtained by AAS (also in Table 2), there is an overestimation of Cu species on the surface, which is unlikely. This behavior suggests that the thermal treatment of activated carbon during the catalyst preparation seems to control the hydrogen adsorption capacity of copper catalyst. Data in Table 2 shows that Cu/CGran (600) and Cu/CGran (750) exhibited the highest values of consumed H_2_, whereas Cu/CGran and Cu/CGran(1000) had the lowest values. Stronger interaction may prevail between the more stable oxygen groups of the carbon support and the copper species, minimizing copper sintering. The presence of oxygenated groups may act as anchors of the copper particles. This is reflected in the observed higher hydrogen adsorption capacity (Table 2), especially for Cu/CGran (600) and Cu/CGran (750) samples, which is consistent with the percent reducibility calculated from TPR measurements. This behavior have attracted substantial interest due to the significant enhancement of hydrogen storage on carbon materials which could improve the reducibility of metal [41]. Most importantly, the observation that hydrogen spillover effect could be enhanced by a variety of functional groups [42,43,44] must also be taken into account. Effectively, Psofogiannakis and Froudakis [45] studied hydrogen spillover effect of metals supported on carbon materials. The authors found that the removal of oxygen-containing acidic functional groups (at lower temperatures) promotes the appearance of hydrogen spillover on metal/activated carbon supported catalysts. Related to the effect on metal active sites, Hu et al. [46] demonstrated quantitatively by chemisorption measurements that an important type of activated Cu sites is enabled by hydrogen spillover. These results are also in agreement with our TPR analysis which suggests that the observed higher copper reducibility is attributed to the selective elimination of acidic oxygen functional groups by thermal treatment. Therefore, the higher reducibility (%) for Cu/CGran (600) and Cu/CGran (750) samples could be related to higher hydrogen adsorption capacity attributed to hydrogen spillover effect.

#### 3.1.7. X-ray Photoelectron Spectroscopy (XPS)

XP spectra of Cu 2p for Cu/CGran and Cu/CGran(750) reduced catalysts and, C 1s XP spectra for Cu/CGran catalyst are shown as reference in Figure 5(a, b, and c, respectively). The XPS results for Cu/CGran and Cu/CGran(x) reduced catalysts, binding energies (eV), Cu/C surface atomic ratios and the Auger αCu parameters, are summarized in Table 3.

Table 3 shows that the binding energy (BE) of the principal component of Cu 2p3/2 is 933.0 ± 0.1 (eV), which can be attributed to Cu^0^ species [47,48,49], suggesting complete reduction of Cu surface species under the experimental conditions used. On the contrary, TPR showed that the reducibility (%) of Cu species was not 100 %. The difference between these two characterization techniques can be attributed to two points: (i) XPS analysis only measures the near-surface of the catalyst; and (ii) TPR measurements detect the Cu oxide species deposited inside the activated carbon porous. Since the binding energies of Cu^0^ and Cu^+^ species are similar, it is difficult to differentiate them based only on Cu 2p3/2. The Auger parameter (αA) can distinguish both species. Based on αA parameter only Cu^0^ species is present on the surface of the catalysts, in agreement with conclusions drawn by a previous work [50]. In addition, Table 3 summarizes three C 1s peaks with binding energies at 284.8, 286.2, and 287.8 ± 0.1 eV, for all catalysts. These BE are assigned to: C-C and/or C=C bonds of aromatic and aliphatic carbon at 284.8 eV [32,51], C–O bonds [52] in phenolic or ether groups at 286.2 eV [53], and C=O bonds in carbonyl groups at 287.8 eV [52].

Table 3 also presents the proportion of the surface C 1s groups (in parenthesis), and shows that they are similar for all catalysts. This suggests that the proportion of oxygen surface groups on the surface of the catalysts are similar after the thermal modification. Table 3 also summarizes the Cu/C surface atomic ratios of all the catalysts. Table 3 shows the apparent dispersion of Cu species on the activated carbon support, expressed as experimental Cu/C atomic ratio obtained from XPS, which is compared to calculated Cu/C atomic ratio estimated from AAS measurement. The experimental value is higher in all the catalysts. This behavior suggests the formation of larger Cu species on the activated carbon surface, or the formation of multilayer Cu species on the outer surface or the mouth of the pores. A similar result was observed previously for Re catalysts supported on activated carbon, which was also attributed to the formation of Re aggregates over a homogeneous monolayer formed on the support surface [34]. On the other hand, Yin-Fan Han et al. [54] have reported by XPS measurements that surface Cu concentrations were much lower than the nominal loadings, suggesting that Cu species are indeed dispersed inside the porous of support. In light of the above considerations, Cu based catalysts supported on CGran (450), CGran (600) and CGran (750), presented lower experimental Cu/C atomic ratio values compared to Cu/CGran and Cu/CGran (1000), and they are also closer to the nominal atomic ratio. These results suggest a more homogeneous surface distribution of Cu species on these three supports, in agreement with their higher percent of reduction. The results are also consistent with the slight decrease of textural properties of the Cu/CGran(x) catalysts compared to the un-modified Cu/CGran catalyst. 

### 3.2. Catalytic Activity of Cu/CGran and CCGran(x) Catalysts

The transformation of glycerol and the evolution of products as a function of time over Cu/CGran and Cu/CGran(x) catalysts are shown in Figure 6. 

The concentration profiles over all the catalysts were similar.1,2-PDO was the main product. Acetol was observed as an intermediate product and 2-propanol was formed as a minor product. Other products observed as trace amounts include propanoic acid, 1,3-PDO, acetic acid, ethylene glycol and other polymerized compounds. The catalytic activity of the catalysts are represented as initial rate expressed in terms of mol of glycerol converted per gram of catalyst per unit time (mol glyc g_cat_
^−1^ s^−1^) and is shown in Figure 7a. Also, the intrinsic rate expressed in terms of molecules of glycerol converted per active site of Cu per unit time (molec glyc AS_Cu_
^−1^ s^−1^) is shown in Figure 7b, and summarized in Table 4. Figure 7a shows an increase in catalytic activity of the Cu/CGran catalysts as the temperature of thermal modification of the support was increased up to 750 °C. This behavior indicates that thermal modification of the carbon support is beneficial to the overall activity by controlling the surface oxygen groups. As previously discussed, the more thermally-stable oxygen enhanced the dispersion of copper, confirmed by the increase in hydrogen adsorption capacity (Table 2) observed especially for Cu/CGran(600) and Cu/CGran(750) samples. On the other hand, when the catalytic activity is expressed in terms of the active site of Cu (intrinsic rate), this trend changes, as observed in Figure 7b.

Figure 7b shows that a similar catalytic activity is observed for all the Cu/CGran(x) catalysts, except for Cu/CGran(1000), where the initial rate decreased drastically compared to the other catalysts. This behavior can be explained by the following: although Cu/CGran(1000) contains the highest amount of Cu surface species, as calculated by N_2_O-TPR (Table 2), the metal active sites exposed are the least active. The results could be associated with the presence of low amount of thermally stable oxygen surface groups on CGran(1000) due to the thermal treatment conditions. This may decrease the interaction with Cu species, and therefore inhibit the hydrogen adsorption capacity, as observed by N_2_O titration (Table 2). It should be noted that the harsh thermal treatment of 1000 °C significantly decreases the apparent surface area (Table 1) which could limit the hydrogen adsorption capacity as shown Table 2.

Table 4 and Figure 8 present the selectivity (%) calculated at 10% of glycerol conversion for all catalytic systems studied. According to literature report [55,56,57,58,59,60,61], hydrogenolysis of glycerol can occur by three typical reaction mechanisms, namely, dehydration-hydrogenation, dehydrogenation-dehydration-hydrogenation, and direct-hydrogenolysis mechanism in which a C–C bond breaking occurs and glycerol is converted by C–C bond cleavage to ethylene glycol and methanol. In the dehydration-hydrogenation mechanism acetol is formed from the dehydration of the –OH linked to the terminal carbon glycerol which is further hydrogenated to 1,2-PDO. 1,2-PDO can also be hydrogenated to 2-propanol [9]. 3-hydroxypropionaldehyde can be formed from the dehydration of the middle –OH of glycerol followed by subsequent hydrogenation to form 1,3-PDO. Propanoic acid and acetic acid can be obtained from consecutive hydrogenation/dehydration of 3-hydroxipropanal [57,58,62].

Figure 8 shows that the copper-based catalysts transformed glycerol mainly to acetol, 1,2-PDO and 2-propanol. In addition, it is observed that as the temperature of the thermal modification of CGran increased from 600 °C to 750 °C higher selectivity to 1,2-PDO is obtained. Cu/CGran (750) exhibited the highest selectivity towards 1,2-PDO. On the other hand Cu/CGran and Cu/CGran (1000) samples present smaller selectivity towards 1,2-PDO compared with the other samples. 

## 4. Discussion

Firstly, the effect of the reducibility (%) of the samples on the observed yield will be discussed in more detail. The influence of the metallic property is remarkable, especially for the Cu/CGran (750) catalyst which presents the highest reducibility with a value of 97% (Table 1). Indeed, the Cu/CGran (750) exhibited the highest 1,2-PDO yield (Figure 6) at about 8 h of reaction. The presence of metallic copper may favor the hydrogenolysis route to form 1,2-PDO. It must be added that the highest hydrogenolytic capacity towards 1,2-PDO by Cu/CGran (750) may be attributed to the positive effect of acidity, conferred by oxygen functional groups present on the activated carbon support. In addition, the accessibility to the copper atoms on this sample which exhibited the highest hydrogen adsorption capacity. These findings suggest that the effect of oxygen surface groups of the support on copper dispersion is an important factor in this reaction. The behavior may be linked to the presence of stable oxygen surface groups such as quinones which enhances interaction of the copper species during the reduction process, leading to catalysts with high hydrogen adsorption capacity (Table 2). In fact, previous works have reported that oxygen groups present at lower temperatures, mainly carboxylic groups, exhibit hydrophobicity, which could hinder the homogeneous distribution of aqueous precursor solutions. On the contrary, hydrophilic groups present at higher temperatures (phenolic, lactonic, and quinonic groups) could facilitate the dispersion of metallic aqueous species on the surface of activated carbon [26,31,34,37]. This is precisely what was observed for the Cu/CGran (450), Cu/CGran (600) and Cu/CGran (750) samples which presented relatively lower Cu/C surface atomic ratios, higher hydrogen adsorption capacity (Table 2), and higher rates initial rates than the un-modified Cu/CGran (Figure 7). This means that the less acidic groups, especially phenolic and quinonic groups are responsible for anchoring Cu species, which minimizes sintering of the copper particles during the decomposition and reduction of the copper nitrate precursor. These results are as expected since mainly quinonic groups are stable under the thermal modification conditions employed for CGran (450), CGran (600), and CGran (750) supports, as deduced by Figure 1. 

Finally, the acid sites present on CGran activated carbon are an important catalytic descriptor in the copper samples, which can also influence the hydrogenolysis of glycerol towards 1,2 PDO production. Previous work has reported that during the formation of the chemisorbed species, the chemical bonds in the glycerol adsorbing molecule are polarized [63]. Therefore, the oxygen species that remained on CGran activated carbon after thermal treatment have an hydrophilic character and acidity, which may polarize the glycerol molecule. Consequently, the acid sites on the activated carbon in contact or neighboring the metallic copper species may favor the dehydration of a terminal OH group of glycerol, promoting the pathway towards acetol formation which is later hydrogenated to 1,2-PDO.

## 5. Conclusions

Glycerol hydrogenolysis to 1,2-propanediol (1,2-PDO) was evaluated over copper supported on activated carbon. The carbon support was thermally modified at different temperatures. The more thermally stable oxygen groups inhibit copper sintering. This was reflected in the observed higher hydrogen adsorption capacity measured by temperature programmed reduction (TPR) of the selectively oxidize surface copper atoms by N_2_O, and also on the initial rates. The thermal treatment of the activated carbon support substantially impacted the apparent surface area, acid properties, and hydrogen adsorption capacity of the copper catalysts. For this reason, the temperature of the thermal treatment was important in the catalytic reactivity. At high temperatures, the interaction of copper species with the carbon support is minimized, leading to large Cu particles, in agreement with TPR results. The results from this study will contribute additional knowledge on the effect of thermal modification on the final properties of the copper-based activated carbon catalyst. The presence of oxygen surface groups with higher thermal stability such as quinones could enhance the interaction of the copper species during the reduction process, and thus generate a final catalyst with high hydrogen adsorption capacity by hydrogen spillover effect.

## Figures and Tables

**Figure 1 materials-13-00603-f001:**
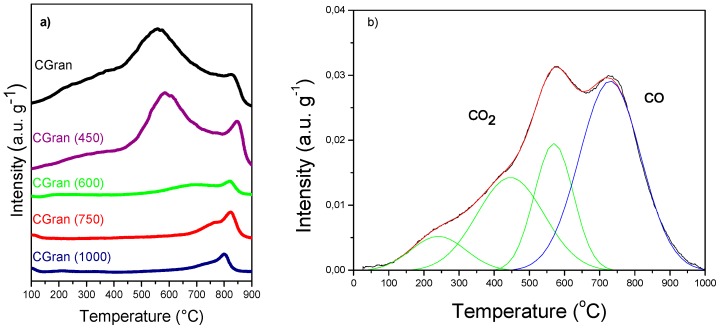
(**a**) Thermal-programmed decomposition (TPD) of CGran and CGran(x) activated carbon supports; (**b**) TPD–MS of un-modified CGran activated carbon.

**Figure 2 materials-13-00603-f002:**
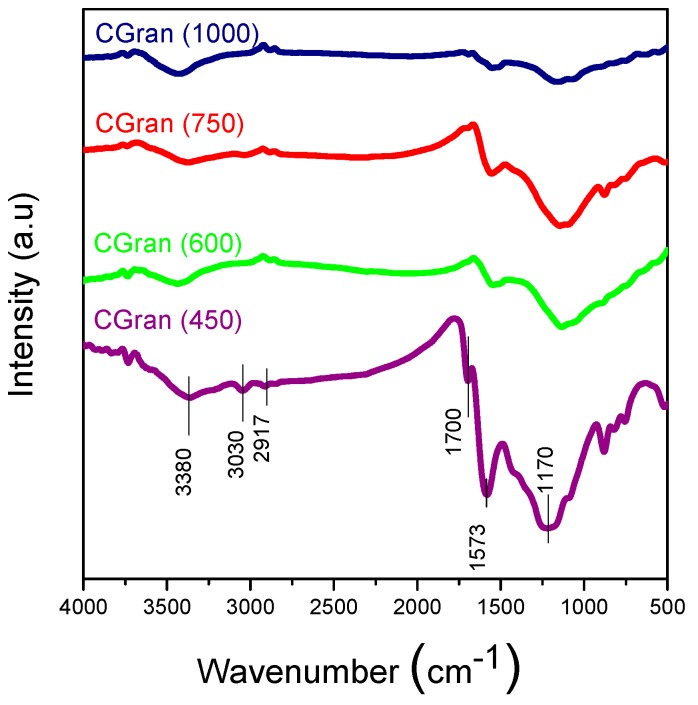
FT-IR of CGran(x) activated carbon supports.

**Figure 3 materials-13-00603-f003:**
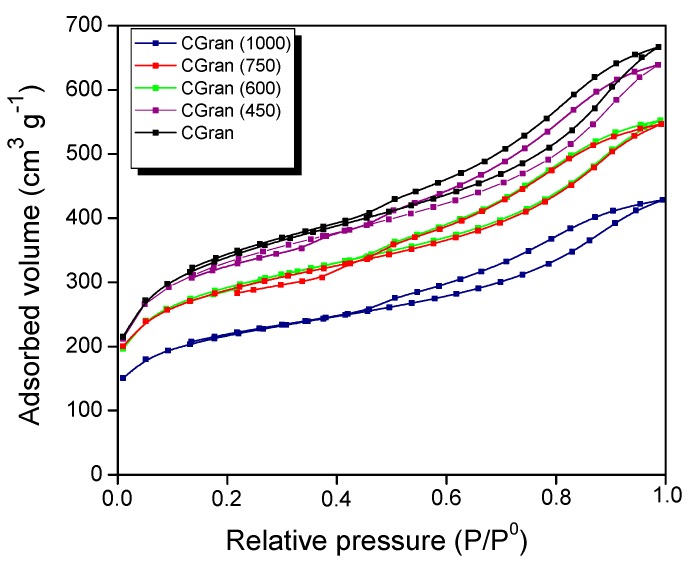
N_2_ adsorption-desorption isotherms at 77 K of CGran and CGran(x) activated carbon supports.

**Figure 4 materials-13-00603-f004:**
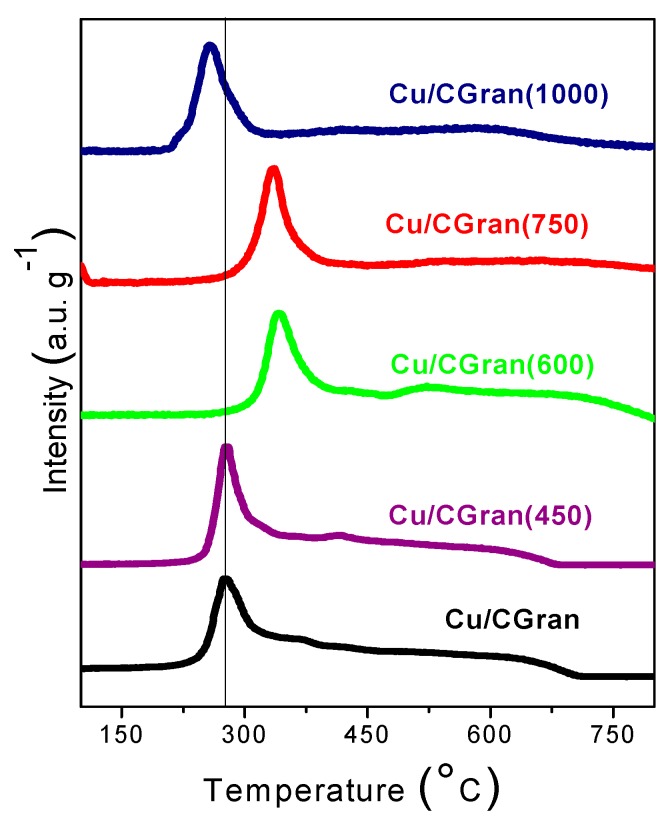
Thermal programmed reduction (TPR) of Cu/CGran and Cu/CGran(x) catalysts.

**Figure 5 materials-13-00603-f005:**
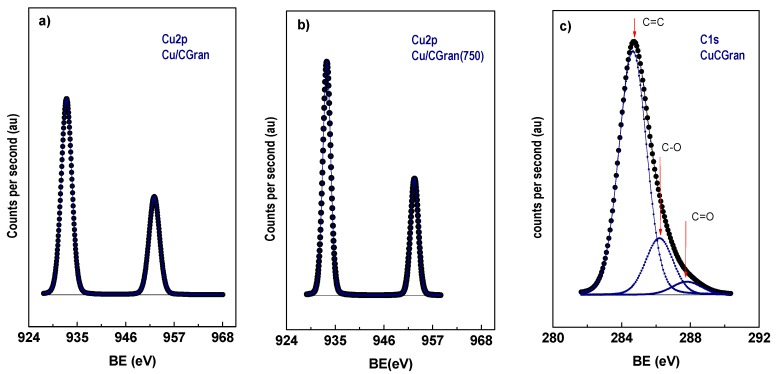
XP spectra of Cu 2p in reduced catalysts (**a**) Cu/CGran, (**b**) Cu/CGran(750), and C1s XP spectra for (**c**) Cu/CGran catalysts.

**Figure 6 materials-13-00603-f006:**
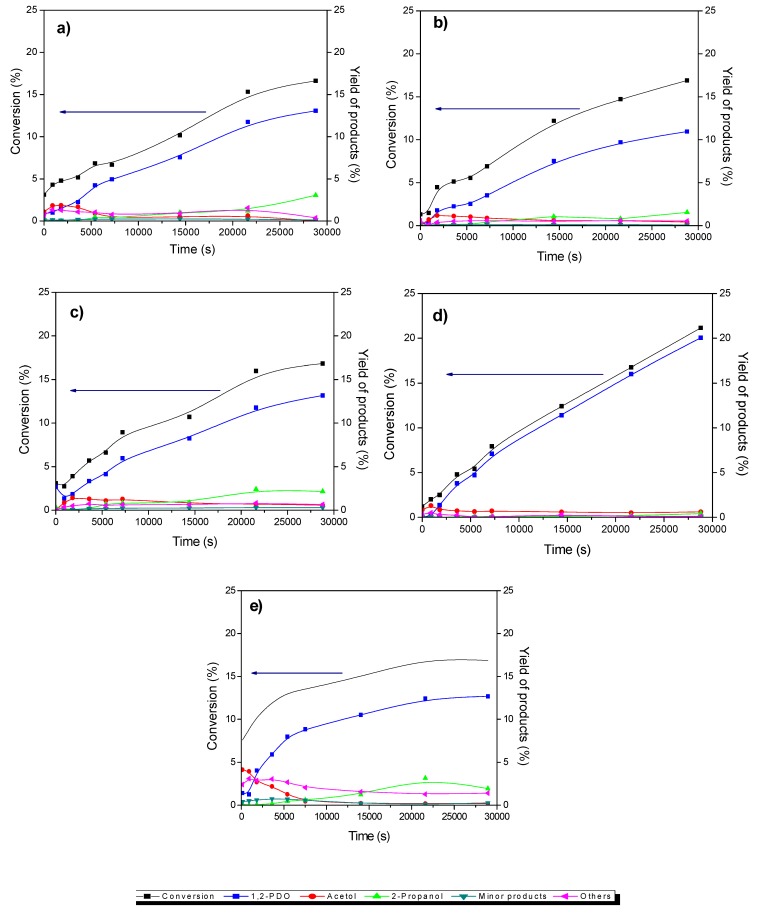
Variation of the transformation of glycerol and the yield of products with time for (**a**) Cu/CGran, (**b**) Cu/CGran(450), (**c**) Cu/CGran(600), (**d**) Cu/CGran(750), (**e**) Cu/CGran(1000) catalysts.

**Figure 7 materials-13-00603-f007:**
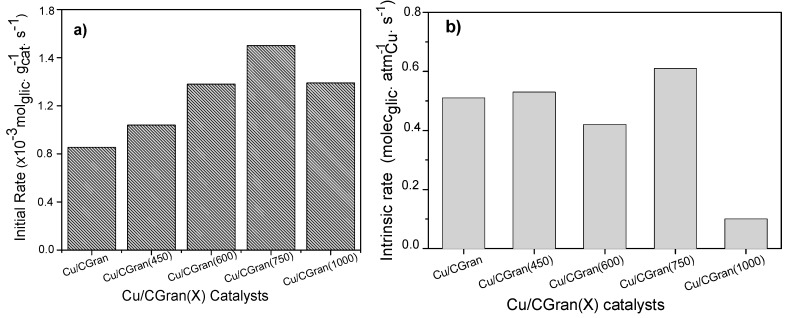
Catalytic activity of Cu/CGran and Cu/CGran(x) catalysts expressed as (**a**) Initial rate (r_0_), (**b**) Intrinsic rate (r_I_).

**Figure 8 materials-13-00603-f008:**
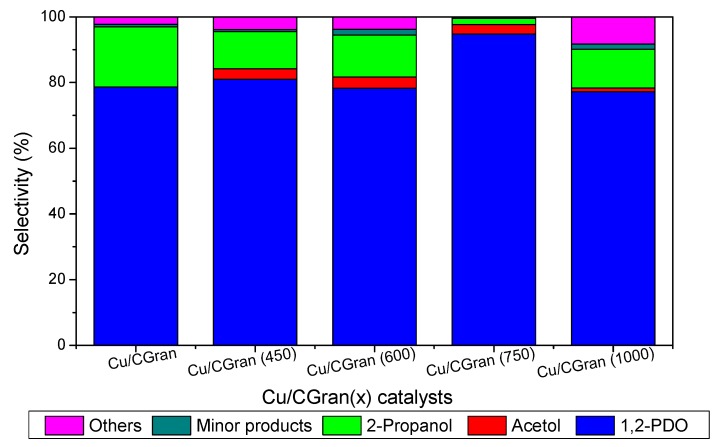
Product distribution calculated at final conversion of glycerol on Cu/CGran and Cu/CGran(x) catalysts.

**Table 1 materials-13-00603-t001:** Physical and chemical properties of the CGran(x) supports and Cu/CGran catalysts.

SUPPORTS							
	Cu Loading (wt%)	S_BET_ (m^2^ g^-1^)	Vp (cm^3^ g^-1^)	Vm (cm^3^ g^-1^)	Vo (cm^3^ g^-1^)	E_0_ (mV)	Reducibility * (%)
CGran	-	1477	1.11	0.72	0.39	171.5	-
CGran (450)		1022	0.95	0.58	0.37		
CGran (600)		893	0.84	0,51	0.33		
CGran (750)		881	0.82	0.50	0.32		
CGran (1000)		667	0.64	0.38	0.26		
CATALYSTS							
Cu/CGran	7.4	559	0.52	0.32	0.20	−134.9	86
Cu/CGran (450)	5.8	687	0.66	0.39	0.27	−140.8	94
Cu/CGran (600)	5.8	652	0.62	0.37	0.25	−144.5	95
Cu/CGran (750)	5.7	579	0.55	0.33	0.22	−149.4	97
Cu/CGran (1000)	6.3	423	0.44	0.24	0.20	−158.3	81

* Calculated from TPR results.

**Table 2 materials-13-00603-t002:** Hydrogen adsorption capacity measured by temperature programmed reduction (TPR) of the selectively oxidize surface copper atoms by N_2_O and initial rates.

Sample	Consumed H_2_ (μmol/gcat)	Cu Active Sites × 10^21^ (N_2_O TPR)	Cu Bulk × 10^20^ (AAS)
Cu/CGran	829	1.00	7.02
Cu/CGran (450)	983	1.18	5.50
Cu/CGran (600)	1635	1.97	5.50
Cu/CGran (750)	1376	1.66	5.41
Cu/CGran (1000)	723	8.71	6.00

**Table 3 materials-13-00603-t003:** Bond energies (eV) and surface atomic ratios of copper carbon species (Cu/C) of Cu/CGran(x) catalysts.

Catalysts	C_1s_ (eV)	Cu2p_3/2_ (eV)	Cu/C (exp.)	Cu/C (nominal)	αCu (eV)
Cu/CGran	284.8 (76) 286.2 (21) 287.8 (3)	933.0	0.022	0.015	1851.1
Cu/CGran (450)	284.8 (77) 286.2 (20) 287.8 (3)	933.0	0.016	0.012	1851.0
Cu/CGran (600)	284.8 (78) 286.2 (20) 287.8 (2)	933.0	0.018	0.012	1851.2
Cu/CGran (750)	284.8 (77) 286.2 (20) 287.8 (3)	933.0	0.017	0.012	1851.0
Cu/CGran (1000)	284.8 (76) 286.2 (20) 287.8 (4)	933.0	0.021	0.013	1850.9

**Table 4 materials-13-00603-t004:** Catalytic activity and selectivity of Cu/CGran(x) catalysts.

Catalysts	Conversion (%)	Initial Rate (×10^−3^ mol_glyc_ g_Cat_^-1^ s^-1^)	Intrinsic Rate (molec_glyc_ AS_Cu_^-1^ s^−1^)	Selectivity (%) (10% of glyc Conversion)				
				1,2-PDO	acetol	2-Pro	Other	Minor Products
Cu/CGran	16.6	0.85	0.51	78.6	0.00	18.4	2.26	0.75
Cu/CGran (450)	13.6	1.04	0.53	81.0	3.13	11.4	3.91	0.60
Cu/CGran (600)	16.8	1.38	0.42	78.3	3.42	12.8	3.82	1.73
Cu/CGran (750)	21.8	1.70	0.61	94.7	2.88	1.91	0.44	0.00
Cu/CGran (1000)	16.9	1.39	0.10	77.2	1.12	11.8	8.57	1.61

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
