# Peer review of "Thermal Modification Effect on Supported Cu-Based Activated Carbon Catalyst in Hydrogenolysis of Glycerol"

_materials, 2020, doi:10.3390/ma13030603_

Round 1

Reviewer 1 Report

The authors attempted to investigate the heat treatment effect on the activated carbon with impregnated copper as the catalyst for glycerol conversion. Carbon supports were treated at various temperatures before metal loading. They expected to study the surface properties affected by heat treatment including surface area, porosity, acidity, and functional groups. However, the techniques and approaches they used failed to provide adequate evidence to support their point of view. Many of the methods they employed were not clearly described and hence caused confusion, as comments 5 and 7. Some significant techniques were missing to study the catalyst features such as SEM (to learn the surface morphology of the catalysts), XRD (to learn the phases present in the catalysts), XPS profiles (Comment 8), etc. The nomination and the description of the catalyst is incorrect, as mentioned in Comment 1. There are also some other minor suggestions or revisions (Comments 2,3,4,6) to improve their work listed at the end of the comments. However, at this point, this manuscript is not recommended for publication in materials.

Comments:

According to the description of the experiment, the material that the authors are interested in this manuscript is copper catalyst supported on activated carbon, carbon as support. But in Abstract section, the author stated the reaction was performed over “copper supported activated carbon”, which conflicts with their design. When DOE is first mentioned in the context, please explain what is the acronym represent for. Introduction section, Page 2, “In this mechanism, the cleavage of C-C bond is proposed to occur through a base-56 catalyzed retro-aldol reaction”. It should be C-O bond. “Therefore, considering the present work evaluates the effect of thermal treatment of activated 92 carbon as support for copper active phase on the glycerol conversion.” Please reword this statement. It is not clear to readers. In Materials and Methods section, the authors claimed that TPD–MS was carried out and CO and CO2 signals were detected by TCD. But in Results section, only one figure (Figure 1) is provided, not specifying what gas signal is this. Where is the result of the other gas? This is very confusing, please explain or add another data. It is obvious that the N2 isotherms in Figure 3 are all Type IV whereas the authors said it is Type 1. The discussion of TPR also leads to confusion. Is it copper or copper oxide that was supported on the activated carbon? If it is copper, how could it be reduced with TPR method? No oxidation procedure was mentioned prior to this experiment. If it is copper oxide, how was it formed? No calcination was mentioned in the experimental section. If it is copper oxide, the title and the nomination of the catalyst should all be revised. If it is copper oxide, N2O could not directly oxidize it. In conclusion, the authors should describe their experimental section and their catalyst in more detail and in a clearer way to inform the readers. Not only table for the peak’s data, XPS profiles should also be provided so that a direct demonstration of the catalyst characterization can be given.

Reviewer 2 Report

Presented paper deals with hydrogenolysis reaction of glycerol to1,2-PDO over copper supported catalysts.Overall text sounds quite good. The characterization apsects are well presented and descirbed (especially XPS) but in my opinion there are not enough methods used to characterize the studied catalysts. SEM/HRTEM should be used in order to collect data of particle size and aggregation/uniformity asepcts. Also cyclic voltammtery (if available) could improve the Cu species identification int ht epresented catalysts ( especially when the carbon is used as a support, carbon is a nice material for the cyclic voltammetry measurements). Why authors did not measure the XRD? According to the catalytic results, why the Authors presented the selectivity results only at conversion of 10%? Did they exclude the diffusion effect?  Did Authors tried another reaction conditions?  The interaction of the copper species during the reduction process thus generating a final catalyst with higher hydrogen adsorption capacity by hydrogen spillover effect could be more detailed discussed in accordance to the available literature data.

Reviewer 3 Report

This manuscript reports the effect of activated carbon heat treatment temperature on the performances of copper/activated carbon for catalyzing glycerol hydrogenolysis to 1,2-propanediol. The subject studied in this manuscript is new because carbon supports are used primarily for noble metals and copper is not a noble metal and carbon-supported catalysts cannot be regenerated. The following information should be included in the manuscript:

The manuscript should explain the advantages of using carbon for supporting copper compared to other supports (reported in references 10-14) because carbon support cannot be regenerated. The manuscript should explain the reason for low glycerol conversion (less than 22%) reported in Table 4 and Figure 5. The manuscript should explain the large differences between CGran data in Table 1 and those reported in reference 34 because CGran used in this manuscript is identical to that used in reference 34. The manuscript should report the reaction conditions (e.g., reaction temperature, catalyst amount, hydrogen pressure) used for obtaining the conversion data in Table 4 and Figure 5. The manuscript should report metal and sulfur contents of the CGran activated carbon because these contents can affect catalytic performances strongly. The manuscript should explain the reason of using high reduction temperature ( line 110) because of the low copper melting temperature and the high reduction temperature might cause serious sintering problem.

Reviewer 4 Report

This work is introduced the supported Cu-based activated carbon catalysts and showing the thermal modification effect on the catalytic performers. Unfortunately, this work could not be accepted at this moment.
Primarily, there is not a great deal of novelty presented here, the Cu-based activated carbon catalysts is not a creative idea by combining with thermal modification effect. Moreover, the characterization part is completely described, especially for the atomic absorption spectroscopy part, which should be important for the present of the Cu-based activated carbon catalysts to show the Cu changed before and after the catalytic process, and how stable for the material. Besides, the authors still need to improve when explaining the Cu loading difference caused by the thermal modification process, as well as the difference before and after the catalytic test. Last, it cannot convince me about the catalytic activity test part. The authors show some results, but more detail should be required to prove the results, such as the more accurate qualitative and quantitative analytical tools that should be used and shown the results here to support the results and conclusion. For these reasons, the manuscript should not be accepted at this moment. However, it may be reconsidered after major revision.

Reviewer 5 Report

The manuscript of Juan Seguel, et al. entitled ”Thermal modification effect on supported Cu-based  activated carbon catalyst in hydrogenolysis of glycerol” deals with a study how the thermal treatment of a carbon support influences on its textural and acidic characteristics, the physico-chemical properties of supported Cu catalysts and their performance in the glycerol hydrogenolysis to 1,2-propanediol. The manuscript is wee-written and contains a lot of interesting results. I’m sure that it will attract a big attention among researchers and help to shed a light on factors influencing the performance of Cu-supported catalysts in hydrogenolysis reactions. I can recommend the manuscript to a publication in the “Materials” provided that my minor comments would be take into account.

1. The experimental, lines 109-110. “Before the reaction, the catalysts were reduced ex-situ at 350ºC for 4 h in flowing H2 (60 mL min-1 110 ).”

After that, according to Ref. [14], the pre-reduced catalyst was added to a reaction mixture in a batch reactor. It means that between these two events the pre-reduced catalyst is occasionally supposed to air and, consequently, could be supposed to a partial oxidation. Are the authors sure that Cu in all catalysts loaded to the batch reactor has zero charge before and during the reaction? According to Fig. 4, Cu reduction may occur at temperature as high as 350 C, while reaction temperature is only 220 C, so an additional Cu reduction during the reaction may not occur. If so, active Cu phase during the reaction could be not only metallic Cu but also Cu2O. Please provide XRD patterns of the pre-reduced Cu/C catalysts under study to prove that no occasional oxidation for these catalysts takes place after the pre-reduction step.

2. The sintering of Cu particles in Cu/C can take place during the reaction, what can influence on the catalytic stability of the catalysts. So, the comparative information on the particle size of Cu species before and after reaction in dependence on the properties of the support would be very desirable.

3. I’m not a big expert in English, but I’d recommend to check the manuscript for English grammar.  

Round 2

Reviewer 4 Report

The author did respond to the comments but still needs to improve the manuscript, especially from Page 8-11 it is very confusing. Could be considered to be published after minor revision.